# Chemical Adsorption Strategy for DMC-MeOH Mixture Separation [note 1]

**DOI:** 10.3390/molecules26061735

**Published:** 2021-03-19

**Authors:** Fucan Zhang, Ping Liu, Kan Zhang, Qing-Wen Song

**Affiliations:** 1State Key Laboratory of Coal Conversion, Institute of Coal Chemistry, Chinese Academy of Sciences, Taiyuan 030001, China; 17865422752@163.com (F.Z.); pingliu@sxicc.ac.cn (P.L.); zhangkan@sxicc.ac.cn (K.Z.); 2State Key Laboratory of Heavy Oil Processing, College of Chemical Engineering and Environment, Chinese China University of Petroleum, Beijing 102249, China

**Keywords:** dimethyl carbonate, superbase, carbon dioxide, reversible adsorption, separation method

## Abstract

The effective separation of dimethyl carbonate (DMC) from its methanol mixture through simple, inexpensive and low energy-input method is a promising and challenging field in the process of organic synthesis. Herein, a reversible adsorption strategy through the assistance of superbase and CO_2_ for DMC/methanol separation at ambient condition was described. The process was demonstrated effectively via the excellent CO_2_ adsorption efficiency. Notably, the protocol was also suitable to other alcohol (i.e., monohydric alcohol, dihydric alcohol, trihydric alcohol) mixtures. The study provided guidance for potential separation of DMC/alcohol mixture in the scale-up production.

## 1. Introduction

Dimethyl carbonate (DMC), defined as a green chemical, has many applications in synthetic chemistry, and it is also used as a solvent, an additive in gasoline, a depurative, and a surfactant [1,2,3]. During the past ten years, much attention has been paid to the carbon dioxide (CO_2_)-based synthetic routes to DMC, such as transesterification of cyclic carbonate and methanol [4,5,6,7] and carbonylation of CO_2_ [8,9,10,11,12]/urea [13] with methanol (Scheme 1). Notably, in these processes, because of the thermodynamic equilibrium and limited catalytic ability, excess methanol was generally used for obtaining the high efficiency. Therefore, the target product was always mixed with methanol, and the easily formed DMC/MeOH binary azeotrope (approximate methanol/DMC composition of 70%:30%) increased the complexity of DMC separation [14,15,16]. Until now, the available separation methods have included extractive distillation, liquid–liquid extraction, selective adsorption, lower temperature crystallization, and the membrane/distillation integrated process, etc. Among them, pressure distillation technique is used in industrial processes, and the energy input is very high. Therefore, improvement and innovation are urgently required for the enhancement of separation efficiency and the decrease in energy consumption.

In 2005, Jessop et al. reported a reversible nonpolar-to-polar solvent [17]. In their work, a non-ionic liquid including an alcohol and an organic base was converted into a salt in liquid form upon exposure to a CO_2_ atmosphere, and then reverted back to its non-ionic form when exposed to nitrogen or argon gas. After this, the reversible process was employed as an effective strategy for CO_2_ capture [18,19,20]. Referring to these studies, we speculated that performing the reversible adsorption strategy with CO_2_, a suitable base and MeOH could also be a potential candidate for the separation of the MeOH/DMC mixture. After the adsorption reaction, DMC stays in the liquid phase and the new formed solid adduct precipitates (Scheme 2). For the mechanism, methanol is initially activated by 1,5-diazabicyclo(4.3.0)non-5-ene (DBN) and subsequently attacks on the CO_2_ molecule with the generation of ionic solid adduct which precipitates from the system.

Herein, a reversible adsorption strategy utilising superbase, CO_2_, and methanol for DMC separation from its methanol mixture at ambient condition was proposed, and the detailed parameters being related to the adsorption efficiency such as component, solvent, temperature and pressure were studied. The work aims to provide a separation protocol featuring as low energy input, renewable, simple and high separation efficiency.

## 2. Results and Discussion

Initially, various bases in combination with stoichiometric methanol were examined on the absorption capacity under the neat conditions (Scheme 3). The conjugated and strong organic bases such as 1,8-diazabicyclo(5.4.0)undec-7-enes (DBU), DBN and 1,1,3,3-tetramethylguanidine (TMG) belonging to superbase [21] showed high absorption ability for 1 h under ambient pressure of CO_2_. In addition, the common bases such as pyridine derivative, *N*,*N*,*N*’,*N*’-tetramethyl-1,6-hexanediamine (TMHDA) and 1-methylimidazole (MeIM) displayed weak absorption ability. The basicity and conjugated structure of base candidates were both key factors for high and fast CO_2_ absorption. Among them, 7.4 mmol CO_2_ was captured through 10 mmol methanol and 10 mmol DBN under identical conditions.

The solvent effect is one of most key factors for CO_2_ capture in the CH_3_OH/DMC mixture, and therefore was investigated using DBN as a base here (Table 1). As seen from the results for the first 1h (entries 1–4), the initial absorption rate decreased sharply. Furthermore, the absorptive amount increased a lot after 16 h. These results reveal that the organic media could reduce the reaction rate, and the reason is probably that the low CO_2_ solubility limited the molecule transfer. In addition, polar solvent is beneficial for the enhancement of CO_2_ absorption rate. To verify these speculations, the solubility of CO_2_ in the solvent used in Table 1 was explored (Appendix A, Appendix A). The order of solubility under ambient conditions is as follows: S(DMF, 58.02 mg/L) > S(DMC, 19.92 mg/L) > S(toluene, 14.58 mg/L) > S(hexane, 6.63 mg/L). The results support the hypothesis that the CO_2_ diffusion process was hindered by the medium. Meanwhile, the trend of the absorption capacity for 16 h was also consistent with the solubility rule (entries 1–4). Moreover, the increase in the temperature could lead to the further decrease in absorption rate (entry 3 vs. 5 and entry 6 vs. 7). Consider the absorption efficiency (entry 6) and operation convenience, the next experiments were carried out at room temperature. With an increase in the pressure, the absorption process was sharply enhanced and saturated absorption loading was achieved even in a shorter time (entries 9–11). Under the pressured conditions, the absorption process also quickly reached the balance for other solvents but with an absorption capacity slightly lower than the maximum absorption capacity (entries 12–14).

The effect of DMC concentration on the absorption was also investigated under atmospheric pressure (Figure 1). With the increase in DMC content in the mixture, the CO_2_ absorption loading decreased quickly during the same time. In total, the absorption efficiency is not enough for application in the separation of CH_3_OH/DMC mixture. Therefore, further improvement was obtained through increasing CO_2_ pressure. Under 1.0 MPa CO_2_ for only 20 min, high CO_2_ absorption capacity was gained (Figure 1b). The separation efficiency of DMC/MeOH is up to 98.7% (Figure 1c).

Here, the freezing-centrifugation technology was used in the separation process. First of all, the adducts are sensitive to the temperature and water, and they will decompose under the heat condition (approximate 35 °C, see entry 7 of Table 1). In addition, in the separation process, one of the most crucial factors is the solubility of the adducts in the DMC or other solvents. The lower solubility will give a higher proportion or purity of DMC in the mother liquid. Taken an example, with the assistance of PhCH_3_, the ratio of DMC to CH_3_OH (most in adduct form) in the mother liquid is up to 20. The DBN, CH_3_OH and CO_2_ from the adduct were successively recovered with heat (> 35 °C). The study provided a new and potential method for DMC and CH_3_OH mixture separation by a low energy-input manner.

In the process of the “two step” method, namely the transesterification of cyclic carbonate (ethylene carbonate or propylene carbonate) and methanol [22], several alcohols coexist, which further increases the complexity of separation. Therefore, typical alcohols including monohydric alcohol, dihydric alcohol, trihydric alcohol and their mixtures were also verified here. As seen in Table 2, both monohydric alcohol and polyhydric alcohol showed high CO_2_ absorption efficiency (entries 1–4). However, with the increase in hydroxyl quantity in alcohol molecule, the absorption rate reduced. The reason is probably that the elevated viscosity hinders CO_2_ molecule transfer. Furthermore, the mixtures of methanol and dihydric alcohols were also demonstrated effectively for CO_2_ absorption with high rate and capacity (entries 5–8).

## 3. Materials and Methods

### 3.1. General Information

General analytic methods. ^1^H-NMR spectra were recorded on 400 MHz spectrometers (Bruker AVANCE IIITM 400 MHz, Baden, Switzerland) using DMSO-*d*_6_ as solvent referenced to DMSO-*d*_6_ (2.50 ppm). ^13^C-NMR was recorded at 100.6 MHz in DMSO-*d*_6_ (39.52 ppm). Multiplets were assigned as singlet, doublet, triplet, doublet of doublet, multiplet, and broad singlet.

### 3.2. Materials

CH_3_OH (analytical reagent, >99.5%, Tianjin Fengchuan Chemistry Reagent Limited Company, Tianjin, China). Carbon dioxide (99.999%, Shanxi Yihong Gas Industry Limited Company, Taiyuan, Shanxi, China). The reagents, i.e., 1,8-diazabicyclo(5.4.0)undec-7-enes (DBU, 99%), 1,5-diazabicyclo(4.3.0)non-5-ene (DBN, 98%), 1,1,3,3-tetramethylguanidine (TMG, 99%), *N*,*N*,*N*′,*N*′-tetramethyl-1,6-hexanediamine (TMHDA, 99%) and 1-methylimidazole (MeIM, 99%), were obtained from Aladdin (Aladdin Industrial Corporation, Shanghai, China) and were used as received.

### 3.3. General Procedure for the Absorption Reaction

The absorption process was performed in a 50 mL autoclave with a glass vessel inside equipped with magnetic stirring under 1.0 MPa CO_2_ (10 mL Schlenk tube under atmospheric pressure of CO_2_). After introducing DBN (10 mmol, 1.24 g), CH_3_OH (10 mmol, 0.32 g), and DMC (0–90 wt%), the autoclave was sealed and filled with CO_2_ to keep the pressure of CO_2_ under 1.0 MPa. Then, the reaction mixture was stirred at 25 °C for 20 min. When the absorption reaction was completed, the residual CO_2_ was carefully released. The absorbed CO_2_ was calculated by the increased weight of the glass vessel. Note that these adducts are very sensitive to the water and temperature. NMR technology was selected for the characterization of these adducts. During the process, trace water molecules were introduced, and a small part of the adducts was decomposed. However, the characterization was not obviously affected, as indicated by the spectrum.

### 3.4. General Procedure for the Separation

Freezing-centrifugation technology was used in the separation process. We took an example with the conditions of DBN (10 mmol, 1.24 g), CH_3_OH (10 mmol, 0.32 g), DMC (27 wt%), and PhCH_3_ (10 mL) under 1.0 MPa CO_2_ at 25 °C for 20 min. After completing the absorption reaction (see Procedure 3.3.), the tube was sealed and stored at −7 °C for one hour, and then centrifuged at 3000 revolutions per minute for one minute. The upper liquid was brought out and analyzed by gas chromatography. The precipitate was decomposed over heat and recovered.

## 4. Conclusions

In conclusion, we have demonstrated an unprecedentedly reversible adsorption strategy using a superbase, CO_2_, and methanol for DMC separation at ambient conditions. The low temperature and viscosity, high CO_2_ pressure, and medium are verified to be beneficial for the adsorption process. This strategy is compatible with a wide range of DMC concentrations, with excellent separation efficiency. The effective separation of DMC from its methanol mixture through simple, inexpensive, and low energy input method is also suitable for other alcohols (i.e., monohydric alcohol, dihydric alcohol, trihydric alcohol) or their mixture. The exploration of the whole separation technology is underway in our laboratory.

## Data Availability

All data included in this study are available upon request by contact with the corresponding author.

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
