# Peer review of "Chemical Adsorption Strategy for DMC-MeOH Mixture Separation†"

_molecules, 2021, doi:10.3390/molecules26061735_

Round 1

Reviewer 1 Report

This is a succinct article explaining a useful methodology. However some aspects need to be clarified. My comments are below.

1) Be careful of the use of phases. Just on the first page, interesterification should be replaced by transesterification, and "till" should be "until".

2) Table 1 is hard to follow. Many entries in the 16 hr column are actually less than 1 hour. How is the pressure recorded and maintained with a balloon of CO2? The discussion refers to the rate of CO2 absorption but no rates are measured. You could say initial absorption is low or high (relative to a solvent-free experiment) but the rate (kinetically speaking) is unclear. The CO2 absorption maximum is 10 mmol, and when you state "With increase of the pressure, the absorption rate was sharply enhanced and saturated absorption was achieved during the shorter time" you actually quote low values. A percentage of maximum absorption would be more intuitive to the reader, and standardized across the text.

3) Figure 1 needs to be organized in a more tidy manner, and please begin y-axis at 0.

4) Can you discuss in more detail the isolation of the adducts and the purity of the recovered dimethyl carbonate? Analysis of the DMC after methanol removal (NMR, GC) would make a stronger argument. Please acquire mass spec for the adducts.

Author Response

Dear reviewer,

Thanks for your time and comments.

The response to the comments, please see the attachment

Best regards,

Qingwen Song

Reviewer 2 Report

This manuscript presents a sound and elegant study focused on the separation of alcohols (in particular methanol) from DMC, using nitrogenated superbases and carbon dioxide. The method is also adequate to capture this latter compoud and, as claimed by the authors, can be performed in low-energy conditions, in contrast to the usual pressure distillation. The paper is very well-written so it is a pleasure to read it.

Regarding the method itself, authors use several solvents as liquid phase and CO2 as gas phase, being part of this compound transferred to the liquid phase to form an solid adduct with methanol (or other alcohol) and the superbase of choice, a solid that precipitates, according to the authors, while DMC remains in the liquid phase. Maybe the authors should include also this information not only in the Introduction section but also in the last part of the Methods section (general procedure...).

Figure 1 quality should be improved, as it seems somewhat blurred. One of the curious facts here is that CO2 absorbs better in a liquid poorer in DMC, why is this so? Moreover, the solvent here is not named, could the authors include CO2 solubility in the solvent and in DMC and discuss on it?

Separation efficiency is very high (higher than 90%), although it is worse is a DMC rich liquid, suggesting that solvents are needed to reduce DMC-methanol interactions (logic, as they form an azeotrope). It would be good that the authors comment on the subsequent separation of DMC from the solvent. Moreover, how the adduct is broken? Can the superbase be recovered and, thus, separated from methanol? Some comments on these subjects will give a more complete vision of the separation process and its value in comparison to the benchmark pressure distillation process.

Author Response

Dear reviewer,

Thanks for your time and comments.

The response to the comments, please see the attachment.

Best regards,

Qingwen Song

Round 2

Reviewer 1 Report

Thank you for considering the recommended changes.